# Multicenter Diagnostic Evaluation of a Novel Coronavirus Antigen Lateral Flow Test among Symptomatic Individuals in Brazil and the United Kingdom

Debora S. Faffe,[a] Rachel L. Byrne,[b] Richard Body,[c,e] Terezinha Marta P. Castiñeiras,[a] Anna P. Castiñeiras,[a] Lorna S. Finch,[b] Konstantina Kontogianni,[b] Daisy Bengey,[b] Rafael Mello Galliez,[a] Orlando C. Ferreira, Jr.,[a] Diana Mariani,[a] Bianca Ortiz da Silva,[a] Sabrina Santana Ribeiro,[a] Margaretha de Vos,[d] LSTM Diagnostic Group,[b] CONDOR Steering Group,[e] Camille Escadafal,[d] Emily R. Adams,[b] Amilcar Tanuri,[a] Ana I. Cubas Atienzar[b]

[a]Universidade Federal do Rio de Janeiro (UFRJ), Núcleo de Enfrentamento e Estudos de Doenças Infecciosas e Emergentes e Reemergentes (NEEDIER), Rio de Janeiro, Brazil

[b]Liverpool School of Tropical Medicine, Centre for Drugs and Diagnostics, Liverpool, United Kingdom

[c]Oxford University, Oxford, United Kingdom

[d]FIND, Geneva, Switzerland

[e]Manchester University NHS Foundation Trust, Manchester, United Kingdom

Debora S. Faffe and Rachel L. Byrne are joint first authors. Author order was decided by reverse alphabetical order. Amilcar Tanuri and Ana I. Cubas Atienzar are joint senior authors.

**ABSTRACT** The COVID-19 pandemic has led to the commercialization of many antigen-based rapid diagnostic tests (Ag-RDTs), requiring independent evaluations. This report describes the clinical evaluation of the Novel Coronavirus 2019-nCoV Antigen Test (Colloidal Gold) (Beijing Hotgen Biotech Co., Ltd.), at two sites within Brazil and one in the United Kingdom. The collected samples (446 nasal swabs from Brazil and 246 nasopharyngeal samples from the UK) were analyzed by the Ag-RDT and compared to reverse transcription-quantitative PCR (RT-qPCR). Analytical evaluation of the Ag-RDT was performed using direct culture supernatants of SARS-CoV-2 strains from the wild-type (B.1), Alpha (B.1.1.7), Delta (B.1.617.2), Gamma (P.1), and Omicron (B.1.1.529) lineages. An overall sensitivity and specificity of 88.2% (95% confidence interval [CI], 81.3 to 93.3) and 100.0% (95% CI, 99.1 to 100.0), respectively, were obtained for the Brazilian and UK cohorts. The analytical limit of detection was determined as $1.0 \times 10^3$ PFU/mL (Alpha), $2.5 \times 10^2$ PFU/mL (Delta), $2.5 \times 10^3$ PFU/mL (Gamma), and $1.0 \times 10^3$ PFU/mL (Omicron), giving a viral copy equivalent of approximately $2.1 \times 10^4$ copies/mL, $9.0 \times 10^5$ copies/mL, $1.7 \times 10^6$ copies/mL, and $1.8 \times 10^5$ copies/mL for the Ag-RDT, respectively. Overall, while a higher sensitivity was claimed by the manufacturers than that found in this study, this evaluation finds that the Ag-RDT meets the WHO minimum performance requirements for sensitivity and specificity of COVID-19 Ag-RDTs. This study illustrates the comparative performance of the Hotgen Ag-RDT across two global settings and considers the different approaches in evaluation methods.

**IMPORTANCE** Since the beginning of the SARS-CoV-2 pandemic, we have witnessed growing numbers of antigen rapid diagnostic tests (Ag-RDTs) being brought to market. In the United Kingdom, this was somewhat controlled indirectly as the government offered free tests from a small number of companies. However, as this has now ceased, individuals are responsible for their own acquisition of test kits. Similarly in Brazil, as of January 2022, pharmacies and other health care retailers are permitted to sell Ag-RDTs directly to the community. Many of these Ag-RDTs have not been externally evaluated, and results are not readily available to the public. Thus, there is now a need for a transparent evaluation of Ag-RDTs with both analytical and

Address correspondence to Ana I. Cubas Atienzar, ana.cubasatienzar@lstmed.ac.uk, or Amilcar Tanuri, atanuri1@gmail.com.

The authors declare a conflict of interest. C.E. and M.d.V. are FIND employees. E.R.A. is employee at Mologic. E.R.A., C.E., and M.d.V. had no role in data collection and analysis. The other authors have no conflicts to declare.

clinical evaluation. We present an independent review of the Novel Coronavirus 2019-nCoV Antigen Test (Colloidal Gold) (Beijing Hotgen Biotech Co., Ltd.), at two sites within Brazil and one in the United Kingdom.

**KEYWORDS** diagnostics, COVID-19, SARS-CoV-2, LFA, Ag-RDT

The COVID-19 pandemic represents the worst health crisis of the last century and has claimed more than 5 million reported lives globally since December 2019 (1), with the true number likely much greater. Despite widespread vaccination programs, SARS-CoV-2 breakthrough infections are commonly reported among the vaccinated. While viral evolution is inevitable and a small number of variants of concern (VOCs) monopolize SARS-CoV-2 transmissibility potential (2), SARS-CoV-2 still poses a significant threat to human life and timely detection of positive cases is vital.

Reverse transcription-quantitative PCR (RT-qPCR) is the gold-standard technique for SARS-CoV-2 diagnosis, and both the United Kingdom and Brazil quickly implemented large-scale RT-qPCR testing for COVID-19 diagnosis such as the National Test and Trace programs implemented in many countries (3). However, throughout the pandemic there have been consistent reports of a delay in results due to slow turnaround times attributed to the transportation of samples to specialized laboratories and sample processing. Additionally, RT-qPCR is expensive and requires sophisticated equipment and highly trained technicians, which prevent its deployment in low-resource and remote settings. Altogether, these limitations hamper timely case isolation and the control of virus transmission locally.

In this scenario, rapid diagnostic tests (RDTs) based on the detection of SARS-CoV-2 antigens (Ag-RDTs) offer a faster and less expensive option for SARS-CoV-2 detection. Their utility has been demonstrated in the United Kingdom, where they were able to expand the number of people tested, thus playing a crucial role in restraining virus transmission (4). Most Ag-RDTs are based on lateral flow immunoassays where a nitro-cellulose membrane presents antibodies that capture viral antigens, usually the SARS-CoV-2 Nucleocapsid (N) protein. There is a myriad of Ag-RDTs on the market; however, their performance varies between brands (5). The WHO published interim guidance in October 2021 highlighting that despite hundreds of Ag-RDT brands available on the market, the number of those examined in published reports is still relatively limited (6). The estimates on performance should also be cautiously interpreted in the context of their methodological limitations and the settings in which they were conducted. The WHO recommends a minimum of 80% sensitivity and 97% specificity, compared to a reference nucleic acid amplification test (NAAT).

This study evaluates the clinical performance of the Coronavirus 2019-nCoV Antigen Test (Colloidal Gold) (Beijing Hotgen Biotech Co., Ltd.), here referred to as Hotgen. The Ag-RDT was evaluated against SARS-CoV-2 RT-qPCR testing in Brazil and the United Kingdom in different settings: a reference center for COVID-19 diagnosis for symptomatic individuals at the Federal University of Rio de Janeiro (Brazil), a community testing clinic in Guapimirim, Rio de Janeiro state (Brazil), and a National Health Service COVID-19 drive-through testing center located in Liverpool (UK).

## RESULTS

**Analytical evaluation.** The limit of detection (LOD) of the Hotgen Ag-RDT was determined as $5.0 \times 10^2$ PFU/mL, $1.0 \times 10^3$ PFU/mL, $2.5 \times 10^2$ PFU/mL, $2.5 \times 10^3$ PFU/mL, and $1.0 \times 10^3$ PFU/mL, giving a viral copy equivalent of approximately $2.1 \times 10^4$ copies/mL, $9.0 \times 10^5$ copies/mL, $1.7 \times 10^6$ copies/mL, $1.8 \times 10^5$ copies/mL, and $8.83 \times 10^5$ copies/mL when tested on the wild-type (WT), Alpha, Delta, Gamma, and Omicron lineages, respectively.

**Clinical evaluation.** The demographics of both the Brazilian and UK study cohorts are shown in Table 1. The median number of days from symptom onset was 3 days (first quartile [Q1] to Q3, 3 to 5), with a median RT-qPCR cycle threshold ($C_T$) value of 21.0 (Q1 to Q3, 17.8 to 24.3) and a complete vaccination level of 34.5% found in Brazil's combined cohorts. When split by testing sites, the community center had a lower

**TABLE 1** Demographics of the Hotgen clinical evaluation cohorts for Brazil and the United Kingdom

| Characteristic[a] | Brazil (total), n = 446 | Brazil reference center, n = 309 | Brazil community center, n = 137 | UK community drive-through, n = 246 |
|---|---|---|---|---|
| Age, yr [median (min−max), N] | 36.0 (18–81), 446 | 35.0 (19–80), 309 | 41.0 (18–81), 137 | 40.6 (18–76), 246 |
| Gender [% F (n/N)] | 59.9% (267/446) | 59.9% (185/309) | 59.9% (82/137) | 57.1% (140/246) |
| Days from symptom onset [median (Q1–Q3); N] | 3 (3–5), 445 | 3 (2–4), 309 | 4 (3–6), 136 | 2 (2–3), 246 |
| Days <0–3 (n, %) | 233, 52.4% | 170, 55.0% | 63, 46.3% | 221, 89.51% |
| Days 4–7 (n, %) | 195, 43.8% | 134, 43.4% | 61, 44.9% | 25, 10.13% |
| Days 8+ (n, %) | 17, 3.8% | 5, 1.6% | 12, 8.8% | 13, 0.45% |
| Vaccinated (n, %) | 154, 34.5% | 126, 40.8% | 28, 20.4% | 194, 78.9% |
| Not vaccinated (n, %) | 292, 65.5% | 183, 59.2% | 109, 79.6% | 51, 20.7% |
| Vaccination not disclosed (n, %) | 0 | 0 | 0 | 1, 0.4% |
| RT-qPCR positivity (%, n/N) | 24.0% (107/446) | 19.4% (60/309) | 34.3% (47/137) | 27.2% (67/246) |

[a]min, minimum; max, maximum; F, female; n, number with characteristic; N, total number.

vaccination rate (20.4% versus 40.8%), older median age (41.0 versus 35.0 years), and higher RT-qPCR positivity rate (34.3% versus 19.4%) than the university center cohort. In the UK cohort, the median number of days from symptom onset was 2 days (Q1 to Q3, 2 to 3), and a complete vaccination level of 78.9% and a median RT-qPCR $C_T$ value of 21.2 (Q1 to Q3, 18.5 to 24.1) were recorded.

In Brazil, 107 of the 446 recruited participants (24.0%) were COVID-19 positive by RT-qPCR (Table 2). From those, 95 (88.8%) were also Ag-RDT positive, while 12 (11.2%) were Ag-RDT negative. No positive Ag-RDT was observed among the 339 individuals with a negative RT-qPCR result. Hotgen Ag-RDT sensitivity and specificity were 88.8% (95% confidence interval [CI], 81.4 to 93.5%) and 100% (95% CI, 98.9 to 100.0%), respectively. Sensitivity was higher in the university center (91.7%; 95% CI, 81.9 to 96.4%) than in the community center (85.1%; 95% CI, 72.3 to 92.6%).

In the United Kingdom, 67 (27.2%) of 246 specimens collected during the enrollment period were positive for COVID-19 by RT-qPCR (Table 2). Fifty-seven of the 67 RT-qPCR-positive individuals (85.1%) were also positive by the Hotgen Ag-RDT, while the remaining 10 were negative. The sensitivity and specificity of the Hotgen Ag-RDT were 85.1% (95% CI, 74.2 to 92.6%) and 100.0% (95% CI, 97.9 to 100.0%) in the United Kingdom.

Both the Brazilian and UK cohorts show that as RT-qPCR $C_T$ value increases, there is a loss of sensitivity of the Hotgen Ag-RDT, with the most optimal sensitivity detected for samples with a $C_T$ value of ≤20.

Subgroup analyses of the Brazilian and UK evaluation cohorts (Table 3) were performed to determine any associated differences in viral antigen detection/sensitivity according to vaccination status or days from symptom onset to test. No discernible differences in sensitivity were detected for recruitment site, participants who were vaccinated compared with those who were unvaccinated, or symptom onset (all P values of >0.05).

## DISCUSSION

The LOD of the Hotgen Ag-RDT met the recommendations, in the WHO target product profile (TPP) for SARS-CoV-2 Ag-RDT, of an acceptable analytical sensitivity/limit of detection at $1.0 \times 10^6$ RNA copies/mL for the WT and Alpha, Delta, and Omicron variants tested, with the Gamma variant slightly outside this threshold (7). The Alpha variant was detected most optimally compared with the Delta and Gamma variants. The Gamma variant was most prevalent in Brazil throughout the recruitment period (8 to 28 June 2021), with a frequency of 100%, while in the United Kingdom, the dominant variant was Alpha, reported at a frequency of 80% at the beginning of the recruitment period (13 May 2021), with Delta rising from 20% to 98% by the end of recruitment (2 July 2021) (8). The LOD for the Omicron lineage was comparable to those for the other lineages, suggesting that sensitivity will not be affected by this new VOC. However,

**TABLE 2** Results and clinical sensitivity and specificity of Hotgen based on COVID-19 RT-qPCR result in Brazil and the United Kingdom[a]

| Result confirmed by RT-qPCR, by location | Result of Hotgen | | | Clinical sensitivity (95% CI), total[b] | $C_T$ % (95% CI), total | | | Clinical specificity (95% CI), total[c] |
|---|---|---|---|---|---|---|---|---|
| | No. with Ag-RDT result | | | | ≤20 | ≤25 | ≤33 | |
| | Positive | Negative | Total | | | | | |
| BR, reference center | | | | 91.7% (81.9, 96.4), 60 | 97.1% (85.5, 99.9), 35 | 94.2% (84.4, 98.4), 52 | 91.7% (81.9, 96.4), 60 | 100.0% (98.5, 100.0), 249 |
| Positive | 55 | 5 | 60 | | | | | |
| Negative | 0 | 249 | 249 | | | | | |
| Total | 55 | 254 | 309 | | | | | |
| BR, community center | | | | 85.1% (72.3, 92.6), 47 | 100.0% (80.7, 100.0), 16 | 97.4% (86.5, 99.9), 38 | 87.0% (91.2, 100.0), 46 | 100.0% (95.9, 100.0), 137 |
| Positive | 40 | 7 | 47 | | | | | |
| Negative | 0 | 90 | 90 | | | | | |
| Total | 40 | 97 | 137 | | | | | |
| UK, community drive-through | | | | 85.1% (74.2, 92.6), 67 | 96.2% (80.3, 99.9), 26 | 83.6% (71.9, 91.8), 61 | 84.84% (73.9, 92.5), 66 | 100.0% (97.9, 100.0), 179 |
| Positive | 57 | 10 | 67 | | | | | |
| Negative | 0 | 179 | 179 | | | | | |
| Total | 57 | 189 | 246 | | | | | |

[a]RT-qPCR, real-time quantitative PCR; $C_T$, cycle threshold; CI, confidence interval; Ag-RDT, antigen-based rapid diagnostic test; BR, Brazil.
[b]Overall clinical sensitivity was 88.2% (95% CI, 81.3, 93.3; total number, 174).
[c]Overall clinical specificity was 100.0% (95% CI, 99.1, 100.0; total number, 518).

**TABLE 3** Hotgen result by onset of symptoms and vaccinated individuals in Brazil and the UK[c]

| Cohort | BR, university center | | | | BR, community center | | | | UK, drive-through | | | |
|---|---|---|---|---|---|---|---|---|---|---|---|---|
| | Ag-RDT positive (n, %) | Ag-RDT negative (n, %) | Sensitivity[a] | 95% CI | Ag-RDT positive (n, %) | Ag-RDT negative (n, %) | Sensitivity[a] | 95% CI | Ag-RDT positive (n, %) | Ag-RDT negative (n, %) | Sensitivity[a] | 95% CI |
| Days from symptom onset | | | | | | | | | | | | |
| Days <0–3 | 28, 16.5% | 142, 83.5% | 93.3% | 76.5, 98.8 | 21, 33.3% | 42, 66.7% | 95.% | 75.1, 99.8 | 48, 24.2% | 150, 75.7% | 87.3% | 75.5, 94.7 |
| Days 4–7 | 26, 19.4% | 108, 80.6% | 92.9% | 75, 98.9 | 17, 27.9% | 44, 72.1% | 77.3% | 54.2, 91.3 | 8, 25.0% | 24, 75.0% | 100% | 63.1, 100.0 |
| Days 8 + | 1, 20.0% | 4, 80.0% | 50.0% | 2.7, 97.3 | 2, 16.7% | 10, 83.3% | 66.7% | 12.5, 98.2 | 0, 0.0% | 13, 100.0% | NA | NA |
| Vaccination received | | | | | | | | | | | | |
| Vaccinated | 18, 14.3% | 108, 85.7% | 85.7% | 62.6, 96.2 | 6, 21.4% | 22, 78.6% | 75.0% | 35.9, 95.5 | 16, 13.2% | 105, 86.8% | 69.6% | 47.1, 86.8 |
| Incomplete vaccination[b] | 22, 21.6% | 80, 78.4% | 100% | 85.1, 100 | 1, 50% | 1, 50% | 100% | 20.7, 100 | 22, 30.1% | 51, 69.9% | 95.6% | 78.1, 99.9 |
| Not vaccinated | 15, 18.5% | 66, 81.5% | 88.2% | 65.7, 96.7 | 33, 30.82% | 74, 69.2% | 86.8% | 72.7, 94.2 | 18, 34.6% | 33, 63.4% | 90% | 68.3, 98.7 |
| Not disclosed | NA | NA | NA | NA | NA | NA | NA | NA | 1, 100.0% | 0, 0.0% | 100.0% | 2.5, 100.0 |

[a]Compared to RT-qPCR.
[b]Single dose received.
[c]CI, confidence interval; Ag-RDT, antigen-based rapid diagnostic test; BR, Brazil; NA, not available.

analytical data should be used with care as a proxy and not to replace clinical evaluation data.

The clinical evaluation utilizing both individual and combined cohorts showed that the sensitivity and specificity of the Hotgen Ag-RDT met the minimum performance requirements outlined by the WHO (6). All sites found that as values for RT-qPCR $C_T$ and median days from symptom onset to test decreased, the sensitivity of the Ag-RDT increased, a finding which is in line with previous evaluations and recommendations of Ag-RDTs to test within 5 days of symptom onset (3, 9). Similarly, the median numbers of days from symptom onset to test were also comparable among overall cohorts. We did not observe a significant difference in diagnostic accuracy among vaccinated, incompletely vaccinated, and SARS-CoV-2 vaccine-naive participants.

Although not statistically significant, the sensitivity obtained in the Brazilian Reference Centre was slightly higher than that in both the community setting in Brazil and the United Kingdom.

The order of the swabs collected from an individual has been shown to influence the sensitivity of test performance (10). In Brazil, the Ag-RDT swab was taken first, followed by the reference swab, with the reverse being true for the United Kingdom. It is likely this reduction of sensitivity observed at the UK site is due to sample depletion. In addition, the two countries used different RT-qPCR assays which targeted separate regions of the SARS-CoV-2 genome, thus potentially influencing the accuracy of the reference standard (11).

The performance of Ag-RDTs has been noted to be affected by the deployment context (12); the diagnostic accuracy in this study could have been influenced by differences in methodology used between settings. For example, in Brazil, Hotgen was performed at the site of sample collection with no delay due to transport to the laboratory compared to the UK cohort. In the United Kingdom, it was not possible to perform the antigen rapid tests at the site due to samples from presumed positive individuals requiring category level 3 (CL3) laboratory processing (13). Additional differences between the Brazilian and UK cohorts include the use of different RT-PCR tests and different swab types collected for both Ag-RDT and RT-PCR testing. It is important to note that despite differences in methods utilized across each site, similar sensitivities and identical values among community settings were obtained.

This study was limited by the exclusion of asymptomatic cases. Asymptomatic individuals pose a significant risk to the control of SARS-CoV-2 (14) and thus require highly sensitive Ag-RDTs. In addition, we were unable to convert the $C_T$ value to viral load across the different RT-qPCR methods. This is not unique to this study, with calls for better standardization of $C_T$ value conversion sounding across the molecular biology community (15).

**Conclusion.** We found the Hotgen Ag-RDT to meet the WHO minimum requirements for Ag-based testing for SARS-CoV-2 and present findings for decision makers to consider its suitability in line with recommended testing strategies, diagnostic capacity, and national testing algorithms in Brazil and the United Kingdom.

## MATERIALS AND METHODS

**Analytical sensitivity.** Viral culture methods to propagate SARS-CoV-2 isolates to assess the limit of detection (LOD) of the Hotgen test followed those previously described elsewhere (16). Briefly, isolates of SARS-CoV-2 from wild-type (REMRQ0001/Human/2020/Liverpool), Alpha (GenBank accession number MW980115), Delta (SARS-CoV-2/human/GBR/Liv_273/2021, GenBank accession number OK392641), Gamma (hCoV-19/Japan/TY7-503/2021), and Omicron (SARS-CoV-2/human/GBR/LIV_1326/2022) lineages were used to evaluate the LOD of the Hotgen Ag-RDT. Frozen aliquots of the third passage of the virus were quantified via plaque assay, and for the determination of the LOD, a fresh aliquot was serially diluted from $1.0 \times 10^6$ PFU/mL. Serial dilutions were directly pipetted into the extraction buffer at a 1/10 ratio for a final concentration of $1.0 \times 10^5$ to $1.0 \times 10^1$ PFU/mL in the extraction buffer. Each dilution was tested in triplicate. Twofold dilutions were made below the 10-fold LOD dilution to confirm the lowest LOD (LLOD). Culture medium was used as the negative control. Viral RNA was extracted from each dilution using the QIAamp viral RNA minikit (Qiagen, Germany) according to the manufacturer's instructions, and genome copy numbers (gcn) per milliliter were quantified using Genesig RT-PCR (Primer Design, UK) as previously described (8).

**Clinical evaluation. (i) Brazil.** Adults with mild COVID-19 symptoms (fever, cough, runny nose, sore throat, anosmia, ageusia, headache, diarrhea, and myalgia) were tested at two diverse settings. The first was a reference center for public health care and security force workers, the COVID-19 Diagnostic Centre (CTD) at the Federal University of Rio de Janeiro, and the second was a community clinic in Guapimirim, a small city in the Rio de Janeiro metropolitan area.

Participants were enrolled from 8 to 28 June 2021. All participants were over 18 years old and signed written informed consent. The study was approved by the National Committee of Research Ethics (CAAE-30161620.0.1001.5257). A questionnaire was applied, including demographic information, symptoms, comorbidities, and exposure risk. Nasal samples were collected for the Ag-RDT according to the manufacturer's instructions and immediately processed at the point of care by trained laboratory researchers. After ~15 min, nasopharyngeal (NP) samples were collected for RT-qPCR from both nostrils using two rayon-tipped swabs. Each swab was left for 15 s, rotated 10 times, and then left for 15 s more. Material was stored in Dulbecco's modified Eagle's medium (DMEM; ThermoFisher Scientific) at 4°C until transportation to the laboratory for RNA extraction. The sample left over was then stored at −80°C for further studies.

Total viral RNA was extracted using the Maxwell 16 viral total nucleic purification kit system (Promega, WI, USA) according to the manufacturer's instructions. Viral RNA was detected using the SARS-CoV-2 (2019-nCoV) CDC qPCR probe assay (Integrated DNA Technologies, IA, USA), targeting the SARS-CoV-2 N1 and N2 genes and the human RNase P gene. All reactions were paired and performed in a 7500 Thermal Cycler (Applied Biosystems, CA, USA). A SARS-CoV-2 RT-qPCR result was considered positive if both targets (N1 and N2) were amplified with a cycle threshold ($C_T$) value of ≤37.

**(ii) United Kingdom.** Adults presenting with symptoms of COVID-19 (including fever, cough, shortness of breath, tight chest, chest pain, runny nose, sore throat, anosmia, ageusia, headache, vomiting, abdominal pain, diarrhea, confusion, rush, or tiredness) at a national testing facility in the community, the Liverpool John Lennon Airport drive-through COVID-19 test center, were asked to participate in the study, and informed consent was obtained. Participants were recruited between 13 May and 2 July 2021 under the Facilitating Accelerated COVID-19 Diagnostics (FALCON) study. Ethical approval was obtained from the National Research Ethics Service and the Health Research Authority (IRAS identifier [ID] 28422, clinical trial ID NCT04408170). A questionnaire was applied, including demographic information and symptoms. For the reference RT-qPCR test, combined throat-nasal (TN) swab samples in universal transport medium (UTM) (Copan Diagnostics Inc., Italy) were used per national standard of care. For the Ag-RDT, nasopharyngeal (NP) swab samples were collected per the manufacturer's instructions. All clinical samples were obtained by health care professionals, transported in cool boxes to the Liverpool School of Tropical Medicine (LSTM), and processed by trained laboratory researchers within 1 to 3 h of collection. Ag-RDTs were performed upon arrival while TN swabs in UTM were aliquoted and stored at −80°C until RNA extraction. RNA was extracted using the QIAamp 96 Virus QIAcube HT kit (Qiagen, Germany) on the QIAcube (Qiagen, Germany), with an internal extraction control inserted at the lysis stage, per the manufacturer's instructions. Samples were screened using the TaqPath COVID-19 (ThermoFisher, UK) assay on the QuantStudio 5 Thermocycler (ThermoFisher, UK). Positive and negative controls were included in each run. The SARS-CoV-2 RT-qPCR result was considered positive if at least two of the three targets (N, ORF1ab, and S) were amplified with a cycle threshold ($C_T$) value of ≤40.

**Statistical analysis.** The sensitivity and specificity, with 95% confidence intervals (CIs), of the Hotgen Ag-RDT were calculated based on the results of the reference method by RT-qPCR assay. Statistical analyses were performed using R scripts and GraphPad Prism 9.1.0 (GraphPad Software, Inc., CA). The 95% confidence interval (CI) for the sensitivity and specificity was calculated using Wilson's test (12). Fisher's exact and chi-square tests were used to determine nonrandom associations between categorical variables.

## ACKNOWLEDGMENTS

We thank all participants who volunteered to take part in the study. In the United Kingdom, special thanks go to the NIHR Clinical Research Network (CRN) for their support with the recruitment, specially to Sue Dowling and Larysa Mashenko for recruitment, sample collection, and processing, and to the CONDOR steering group—A. Joy Allen, Julian Braybrook, Peter Buckle, Paul Dark, Kerrie Davis, Adam Gordon, Anna Halstead, Charlotte Harden, Colette Inkson, Naoko Jones, William Jones, Dan Lasserson, Joseph Lee, Clare Lendrem, Andrew Lewington, Mary Logan, Massimo Micocci, Brian Nicholson, Rafael Perera-Salazar, Graham Prestwich, D. Ashley Price, Charles Reynard, Beverley Riley, John Simpson, Valerie Tate, Philip Turner, Mark Wilcox, and Melody Zhifang—for oversight of the trial in the United Kingdom.

LSTM Diagnostic group: Daisy Bengey, Kate Buist, Karina Clerkin, Thomas Edwards, Jahanara Wardale, Christopher T. Williams, and Dominic Wooding.

This work was funded as part of FIND's work as a coconvener of the diagnostics pillar of the Access to COVID-19 Tools (ACT) Accelerator, including support from Unitaid (grant number 2019-32-FIND MDR), the government of the Netherlands (grant number MINBUZA-2020.961444), and the UK Department for International Development (grant number 300341-102).

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
