## [Reviewer comments · Microbiology Spectrum]

Microbiology Spectrum

Multicentre diagnostic evaluation of a novel coronavirus antigen lateral flow test among symptomatic individuals in Brazil and The United Kingdom

Débora Faffe, Rachel Byrne, Richard Body, Terezinha Castiñeiras, Ana Carla Castiñeiras, Lorna Finch, Konstantina Kontogianni, Daisy Benguey, Rafael Galliez, Orlando Ferreira Jr, Diana Mariani, Bianca da Silva, Sabrina Ribeiro, Margaretha de Vos, Camille Escadafal, Emily Adams, Amilcar Tanuri, and Ana Cubas-Atienzar

Corresponding Author(s): Ana Cubas-Atienzar, Liverpool School of Tropical Medicine

Review Timeline:

Submission Date:	August 4, 2022
Editorial Decision:	October 4, 2022
Revision Received:	October 20, 2022
Accepted:	October 31, 2022

Editor: Rosemary She

Reviewer(s): Disclosure of reviewer identity is with reference to reviewer comments included in decision letter(s). The following individuals involved in review of your submission have agreed to reveal their identity: Priyanka Kapoor (Reviewer #1)

Transaction Report:

DOI: <https://doi.org/10.1128/spectrum.02012-22>

October 4, 2022

Dr. Ana I. Cubas-Atienzar
Liverpool School of Tropical Medicine
Liverpool
United Kingdom

Re: Spectrum02012-22 (Multicentre diagnostic evaluation of a novel coronavirus antigen lateral flow test among symptomatic individuals in Brazil and The United Kingdom)

Dear Dr. Ana I. Cubas-Atienzar:

Thank you for submitting your manuscript to Microbiology Spectrum. It has been reviewed by two experts in the field and the consensus decision is Modifications. When submitting the revised version of your paper, please provide (1) point-by-point responses to the issues raised by the reviewers as file type "Response to Reviewers," not in your cover letter, and (2) a PDF file that indicates the changes from the original submission (by highlighting or underlining the changes) as file type "Marked Up Manuscript - For Review Only". Please use this link to submit your revised manuscript - we strongly recommend that you submit your paper within the next 60 days or reach out to me. Detailed instructions on submitting your revised paper are below.

Link Not Available

Sincerely,

Rosemary She

Journals Department
Reviewer comments:

Reviewer #1 (Comments for the Author):

Suggestions:

1. Abstract

The line is unclear for understanding. Kindly modify it or make it into two sentences for better comprehension. "The analytical limit of detection was determined at 1.0×10^3 pfu/mL, 2.5×10^2 pfu/mL, 2.5×10^3 pfu/mL, and 1.0×10^3 36 pfu/ml giving a viral copy equivalent of approximately 2.1×10^4 copies/mL, 9.0×10^5 copies/mL, 1.7×10^6 37 copies/mL, and 1.8×10^5 38 for the Ag-RDT, when tested on the Alpha (B 1.1.7), Delta (B 1.617.2), Gamma (P.1),³⁹ and Omicron (B.1.1.529) variant, respectively. "

2. Is there any clinical implication of a fall in sensitivity with a rising Ct value? what are the author's interpretations?

3. Any recommendations that can be proposed by authors for future trials for Antigen kit testing

Reviewer #2 (Comments for the Author):

Excellent study comparing the Hotgen Ag-RDT performance in different contexts.

My point:

- "The LOD using Omicron lineage was comparable to the other lineages, suggesting that sensitivity will not be affected by this new VOC."

The omicron variant may have N- or S-gene mutations which may affect the sensitivity of antigen tests. Due to that, some tests had to be modified to detect omicron. Could you discuss about the most frequent mutations of the omicron variant and how they can affect the sensitivity of this specific antigen test?

<https://www.fda.gov/medical-devices/coronavirus-covid-19-and-medical-devices/sars-cov-2-viral-mutations-impact-covid-19-tests>

Staff Comments:

Preparing Revision Guidelines

Please return the manuscript within 60 days; if you cannot complete the modification within this time period, please contact me. If you do not wish to modify the manuscript and prefer to submit it to another journal, please notify me of your decision immediately so that the manuscript may be formally withdrawn from consideration by Microbiology Spectrum.

Rachel Louise Byrne
Liverpool School of Tropical Medicine
Pembroke Place
Liverpool
L5 3QA

Rachel.byrne@lstmed.ac.uk

Manuscript ID: Spectrum02012-22

Dear Dr She,

Thank you for allowing us to submit a revision of our manuscript entitled "*Multicentre diagnostic evaluation of a novel coronavirus antigen lateral flow test among symptomatic individuals in Brazil and The United Kingdom*". We appreciate the careful review and constructive suggestions. We thank the referees for their very helpful comments that have improved our manuscript.

Please find below our response to all comments. For clarity, reviewer comments are in black and ours are consistently in red. The revision has been developed in consultation with all coauthors, and each author has given approval to the final form of this revision.

Thank you for your consideration, please don't hesitate to contact me if you require any clarification on the edits made.

Sincerely,

Rachel Byrne

Suggestions:

1. Abstract

The line is unclear for understanding. Kindly modify it or make it into two sentences for better comprehension. "The analytical limit of detection was determined at 1.0×10^3 pfu/mL, 2.5×10^2 pfu/mL, 2.5×10^3 pfu/mL, and 1.0×10^3 36 pfu/ml giving a viral copy equivalent of approximately 2.1×10^4 copies/mL, 9.0×10^5 copies/mL, 1.7×10^6 37 copies/mL, and 1.8×10^5 38 for the Ag-RDT, when tested on the Alpha (B 1.1.7), Delta (B 1.617.2), Gamma (P.1),³⁹ and Omicron (B.1.1.529) variant, respectively. "

Thank you for this helpful comment. We have now restructured the sentence to "The analytical limit of detection was determined at 1.0×10^3 pfu/mL (Alpha (B 1.1.7)), 2.5×10^2 pfu/mL (Delta (B 1.617.2)), 2.5×10^3 pfu/mL (Gamma (P.1)), and 1.0×10^3 pfu/ml Omicron (B.1.1.529) giving a viral copy equivalent of approximately 2.1×10^4 copies/mL, 9.0×10^5 copies/mL, 1.7×10^6 copies/mL, and 1.8×10^5 for the Ag-RDT respectively."

2. Is there any clinical implication of a fall in sensitivity with a rising Ct value? what are the author's interpretations?

A very good question. We are not clinicians so have not commented on the clinical implications for this study and focused rather on at home or self-testing. A rising Ct value would indicate a low viral load thus a fall in sensitivity would be expected but also necessary. A very sensitive test past the level of clinical infection (individuals capable of transmitting the virus) could result in individuals missing workdays/quarantining when it's not necessary.

3. Any recommendations that can be proposed by authors for future trials for Antigen kit testing
We feel this was addressed in our introduction where we state the need for clinical evaluation in an independent setting. "There is a myriad of Ag-RDTs in the market, however their performance varies between brands [5]. The WHO published interim guidance in October 2021 highlighting that despite hundreds of Ag-RDTs test brands available on the market, the number of those examined in published reports is still relatively limited [6]. The estimates on performance should also be cautiously interpreted in the context of their methodological limitations and the settings in which they were conducted."

Reviewer #2 (Comments for the Author):

Excellent study comparing the Hotgen Ag-RDT performance in different contexts.

My point:

- "The LOD using Omicron lineage was comparable to the other lineages, suggesting that sensitivity will not be affected by this new VOC."

The omicron variant may have N- or S-gene mutations which may affect the sensitivity of antigen tests. Due to that, some tests had to be modified to detect omicron. Could you discuss about the most frequent mutations of the omicron variant and how they can affect the sensitivity of this specific antigen test?

<https://www.fda.gov/medical-devices/coronavirus-covid-19-and-medical-devices/sars-cov-2-viral-mutations-impact-covid-19-tests>

Whilst reviewer two makes an excellent point we feel this falls outside the remit of this paper. We acknowledge other manufacturers have had to make adaptations, but our evaluation is only of Hotgen. We included all lineages to evaluate its use in variants of concerns to which we found no statistical difference.

We would support a review paper discussing these limitations and offering considerations to evaluation sites to conduct robust studies.

October 31, 2022

Dr. Ana I. Cubas-Atienzar
Liverpool School of Tropical Medicine
Liverpool
United Kingdom

Re: Spectrum02012-22R1 (Multicentre diagnostic evaluation of a novel coronavirus antigen lateral flow test among symptomatic individuals in Brazil and The United Kingdom)

Dear Dr. Ana I. Cubas-Atienzar:

Your manuscript has been accepted, and I am forwarding it to the ASM Journals Department for publication. You will be notified when your proofs are ready to be viewed.

Sincerely,

Rosemary She
Editor, Microbiology Spectrum
